# Actomyosin and CSI1/POM2 cooperate to deliver cellulose synthase from Golgi to cortical microtubules in Arabidopsis

Lu Liu[1,5], Ting Wang[1,5], Yifan Bai[1], Pengcheng Yan[1], Liufeng Dai[1], Pingzhou Du[2], Staffan Persson[3,4] & Yi Zhang [1]✉

As one of the major components of plant cell walls, cellulose is crucial for plant growth and development. Cellulose is synthesized by cellulose synthase (CesA) complexes (CSCs), which are trafficked and delivered from the Golgi apparatus to the plasma membrane. How CesAs are released from Golgi remains largely unclear. In this study, we observed that STELLO (STL) family proteins localized at a group of small CesA-containing compartments called Small CesA compartments (SmaCCs) or microtubule-associated CesA compartments (MASCs). The STL-labeled SmaCCs/MASCs were directly derived from Golgi through a membrane-stretching process: membrane-patches of Golgi attached to cortical microtubules, which led to emergence of membrane-tails that finally ruptured to generate SmaCCs/MASCs associated with the cortical microtubules. While myosin propelled the movement of Golgi along actin filaments to stretch the tails, the CesA-microtubule linker protein, CSI1/POM2 was indispensable for the tight anchor of the membrane-tail ends at cortical microtubules. Together, our data reveal a non-canonical delivery route to the plasma membrane of a major enzyme complex in plant biology.

Plant cells are enclosed by strong yet pliant cell walls, which allow plant cells to adhere to each other, define growth and morphology of plant cells, provide mechanical support to constrain turgor pressure and protect cells against pathogens[1]. Cellulose is a prominent cell wall component and consists of long load-bearing and crystalline microfibrils that dictate mechanical properties of the cell wall. Unlike most other cell wall polysaccharides, which are synthesized in the Golgi apparatus before secretion to the apoplast, cellulose polymers are de novo produced at the cell surface by plasma membrane (PM)-localized cellulose synthase (CesA) complexes (CSCs)[2–5]. The CSCs are thought to be heterotrimeric and contain 18 CesA subunits[6,7]. For example, CesA1, CesA3 and CesA6-like proteins (CesA2, CesA5, CesA6 and

CesA9) form primary wall CSCs and CesA4, CesA7 and CesA8 are involved in secondary wall cellulose synthesis in Arabidopsis[8–10].

The CesAs are made at the endoplasmic reticulum (ER) and are thought to assemble into CSCs in the Golgi apparatus, where CSCs have been observed as sixfold symmetrical rosettes by freeze-fracture transmission electron microscopy[11–13]. The assembled CSCs are then delivered to the PM via the trans-Golgi Network (TGN). Here, they get activated and begin to synthesize cellulose. The cellulose microfibrils become immobilized during the incorporation into the cell wall and further synthesis therefore pushes the CSC forward in the PM[5]. The movement occurs bidirectionally along linear tracks that coincide with cortical microtubules tethered to the PM[2,3]. The connection between

[1]Key Laboratory of Cell Proliferation and Regulation Biology of Ministry of Education, College of Life Science, Beijing Normal University, 100875 Beijing, China. [2]Key Laboratory of Cell Proliferation and Regulation Biology of Ministry of Education, Instrumentation and Service Center for Science and Technology, Beijing Normal University, 519087 Zhuhai, China. [3]Copenhagen Plant Science Center (CPSC), Department of Plant & Environmental Sciences, University of Copenhagen, 1871 Frederiksberg C, Denmark. [4]Joint International Research Laboratory of Metabolic & Developmental Sciences, State Key Laboratory of Hybrid Rice, SJTU-University of Adelaide Joint Centre for Agriculture and Health, School of Life Sciences and Biotechnology, Shanghai Jiao Tong University, Shanghai, China. [5]These authors contributed equally: Lu Liu, Ting Wang. ✉e-mail: yi.zhang@bnu.edu.cn

CesAs and microtubules is largely mediated by CESA INTERACTIVE PROTEIN 1 (CSI1), which is also called POM-POM2 (POM2)[14–16]. CSI1/POM2 can bind both CesAs and microtubules, and CesAs fail to track along cortical microtubules in *csi1/pom2* mutants[15–17].

The assembly and delivery of the CSCs to the PM is critical for cellulose biosynthesis in plants[4,13,18–22]. Several factors that regulate the intracellular trafficking and exocytosis of CSCs are identified. For example, two putative glycosyltransferases, STELLO (STL) 1 and STL2, are required for the assembly of CSCs in *Arabidopsis*[23]. Both STL proteins localize at Golgi, where they interact with the CesAs. Loss of STL functions leads to a range of defects in the CesA behavior, including abnormal spatial distribution in the Golgi, reduced secretion to the PM and decreased speed at the PM[23]. Apart from these proteins, malfunctions of the general machinery of endosomal trafficking and exocytosis, such as the pH homeostasis within TGN/early endosome (TGN/EE)[24], the exocyst complex and PATROL1[25], and actomyosin[26–28], negatively impact cellulose synthesis. In particular, the exocytosis of CSCs is mediated by close cooperation between the exocyst complex, PATROL1 and CSI1/POM2[25]. CSI1/POM2 marks the vesicle docking site on cortical microtubules to define the exact position for CSC secretion[25]. PATROL1 interacts with CSI1/POM2 and exocyst subunits and facilitates the tethering/fusion of CSC-containing vesicles to the PM[25]. Furthermore, myosin XIK interacts with the exocyst complex to facilitate the tethering of CSC-containing vesicles to the PM[28]. Several other proteins appear to be associated with secretion or recycling of the CSCs, including the domain of unknown function and plant specific protein, TRANVIA (TVA)[29], the G-protein-coupled receptor-like proteins 7TM1 and 7TM5[30], and a pair of PM-localized proteins that negatively regulate CSC exocytosis, SHOU4 and SHOU4L[31]. In *tva* mutants, the CSC secretion to, and dynamics and density at the PM are disturbed, which led to reduced cellulose content and enhanced sensitivity to cellulose synthesis inhibitors[29]. The partial colocalization of TVA with CesAs at TGN and small CesA-containing compartments suggests that TVA may facilitate the exit of CSCs from TGN and/or the delivery of CSCs to the PM[29]. While these reports provide exquisite insights into the trafficking and exocytosis of CSCs, how the CSCs exit the Golgi is less well researched.

Live-cell imaging of fluorescently labeled CesAs revealed localization of CesAs in post-Golgi vesicles referred to as small CesA compartments (SmaCCs) or microtubule-associated CesA compartments (MASCs), along with localization in Golgi, the PM and occasionally TGN[26,32,33]. SmaCCs/MASCs mainly exist at the cell cortex, where they typically colocalize with cortical microtubules and move processively with the depolymerizing microtubule ends[32,33]. Osmotic stress or other treatments that inhibit cellulose synthesis cause a rapid decrease of CSCs at the PM and a simultaneous increase of SmaCCs/MASCs, indicating that at least a portion of SmaCCs/MASCs come from CSC internalization[32,33]. CSI1/POM2 colocalizes with CesAs at SmaCCs/MASCs and is required for the formation of stress-induced SmaCCs/MASCs[34]. The formation of SmaCCs/MASCs is, furthermore, critical for fast recycling of CSCs to the PM during recovery from the stress, as SmaCCs/MASCs then deliver CSCs to the PM[34]. SmaCCs/MASCs also exist in plant cells grown under non-stressed conditions, when they are most frequently observed in fully elongated cells[32]. Interestingly, some Golgi bodies are observed to pause and associate with the SmaCCs/MASCs under non-stressed conditions[32,33]. However, it is unclear how these SmaCCs/MASCs are formed and how the connection between Golgi and SmaCCs/MASCs is established.

In this study, we observe that STLs label a group of SmaCCs/MASCs that are directly derived from Golgi. By taking advantage of this labelling, we demonstrate that actomyosin and CSI1/POM2 cooperate to promote the formation of the Golgi-derived SmaCCs/MASCs, which represents a non-canonical route for CSC transfer from Golgi to cortical microtubules.

# Results

## STL1 and STL2 localize to Golgi-derived SmaCCs/MASCs

To determine the behavior of the STL proteins in different cells and under different growth conditions, we examined the subcellular localization of the proteins using plants expressing mCherry-labeled STL1 (STL1-mCherry) under the control of the ubiquitin10 promoter, as well as EGFP-tagged STL2 (GFP-STL2) under the control of *STL2* endogenous promoter in the *stl1 stl2* double mutant background. Both fusion proteins were functional as the constructs complemented the *stl1 stl2* phenotype, respectively (Supplementary Fig. 1)[23]. We imaged elongating epidermal cells in etiolated hypocotyls, which are used as a cell model to observe CesA dynamics[2,32,33]. In these cells, STL1-mCherry and GFP-STL2 were mainly detected as ring-shaped compartments that were previously shown to be the Golgi apparatus[23]. Interestingly, both STLs were occasionally observed at small intracellular compartments (Supplementary Fig. 2a). This localization became increasingly evident in epidermal cells at the basal region of etiolated hypocotyls where growth had ceased (Fig. 1a). The small compartments sometimes bifurcated or merged (Fig. 1b, c), mimicking the behavior of SmaCCs/MASCs[15,33].

To determine if the small compartments were SmaCCs/MASCs, we performed colocalization analysis of GFP-STL2 with mCherry-TUA5 and tdTomato-CesA6, respectively. GFP-STL2-labeled small compartments coincided with cortical microtubules and tracked with depolymerizing microtubule ends (Fig. 1d, e and Supplementary Movie 1). GFP-STL2 also co-localized with tdTomato-CesA6 in the small compartments (Fig. 1f). The two proteins co-migrated and moved in an erratic and irregular way (Fig. 1g, h), characteristic of the movement of SmaCCs/MASCs[15,33]. These data corroborated that GFP-STL2-labeled small compartments were SmaCCs/MASCs. Interestingly, GFP-STL2 did not label all SmaCCs/MASCs, but only around 43% of them ($n = 205/473$ in 10 cells; Fig. 1f). Indeed, when comparing fluorescence coincidence between tdTomato-CesA6 and GFP-STL2 we found that the GFP and tdTomato signals overlapped each other to a similar level at Golgi-like compartment, but that the GFP signal covered significantly less of the tdTomato signal at the SmaCCs/MASCs (Fig. 1i). These data imply that STLs do not label all SmaCCs/MASCs, but only a subgroup of them.

We next determined through which route STL-labeled SmaCCs/MASCs were generated. As indicated above, we rarely detected SmaCCs/MASCs under non-stressed conditions in rapidly elongating cells[33]. However, application of the cellulose synthase inhibitor isoxaben induced CSC internalization from the PM into SmaCCs/MASCs (Supplementary Fig. 2b)[33]. Interestingly, GFP-STL2 did not co-localize with these compartments (Supplementary Fig. 2b, c). Instead, time-lapse imaging of basal hypocotyl cells, where SmaCCs/MASCs were abundant, revealed that GFP-STL2-labeled SmaCCs/MASCs were directly derived from Golgi through a "membrane tail-stretching" process. Here, parts of the Golgi membrane protruded and formed a long tail, which finally delivered a cortical SmaCC/MASC through membrane fission (Fig. 1j, k and Supplementary Movie 2). Strikingly, we did not observe similar Golgi membrane tail-stretching events for the Golgi marker SYP32-mCherry (Supplementary Fig. 3a and Supplementary Movie 3). Moreover, the TGN marker SYP61-CFP did not colocalize with GFP-STL2 in the Golgi tail nor the SmaCCs/MASCs (Supplementary Fig. 3b–d and Supplementary Movie 4). These observations indicated that the Golgi membrane tail-stretching events represented a non-canonical route to form SmaCCs/MASCs. Together, these data supported a heterogenous population of SmaCCs/MASCs and highlight the STLs as markers for Golgi-derived SmaCCs/MASCs.

We next asked if the Golgi membrane tail-stretching events also exist in other cell types and under different growth conditions. For this purpose, the localization of GFP-STL2 was examined in petiole epidermal cells of light-grown seedlings. Similar to our observations in the

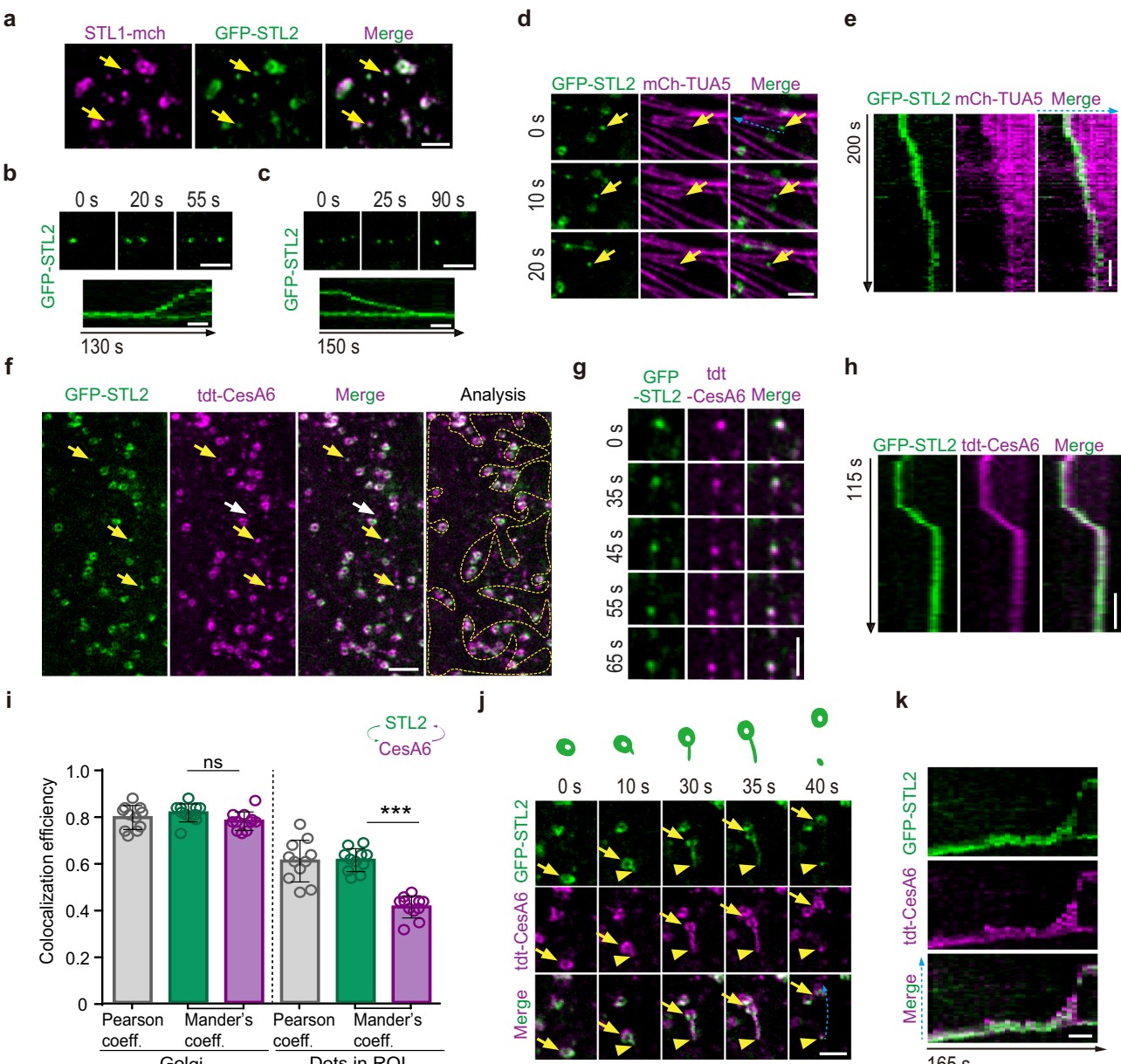

**Fig. 1 | STL1 and STL2 localized to the SmaCCs/MASCs that were derived from Golgi. a** Representative images of an epidermal cell expressing both STL1-mCherry and GFP-STL2 in the basal region of 4-day-old etiolated hypocotyls. Arrows indicated colocalization at the small compartments. Bar = 2 μm. **b, c** Time-lapse images and kymograph analysis showing the split of GFP-STL2 signal into two (**b**) and the merge of two compartments into one (**c**). Bars = 2 μm. **d, e** Time-lapse images (**d**) and kymograph analysis (**e**) showing that GFP-STL2 tracked the depolymerizing ends of microtubules. Bars = 2 μm. **f** Representative images showing partial co-localization of tdTomato (tdt)-CesA6 and GFP-STL2 in the epidermal cells in the basal region of 4-day-old etiolated hypocotyls. Yellow arrows indicated the SmaCCs/MASCs containing both tdt-CesA6 and GFP-STL2, while white arrows indicated SmaCCs/MASCs without GFP-STL2. For analysis, region of interest (ROI) that excluded the Golgi apparatuses was marked by yellow dotted outlines. Bar = 3

μm. **g, h** Time-lapse images (**g**) and kymograph analysis (**h**) showing that tdt-CesA6 and GFP-STL2 co-migrated at SmaCCs/MASCs. Bars = 2 μm. **i** Colocalization analysis of GFP-STL2 and tdt-CesA6 in the Golgi and ROI as indicated in (**f**), using the Pearson correlation coefficient and Mander's coefficient. The arrow schemes indicated the intensity overlap of GFP-STL2 with tdt-CesA6 (green) and the reversals (magenta). Values are mean ± SD. *n* = 11 cells from 5 seedlings; ***P value < 0.001; ns, not significant; two-sided Student's *t* test. **j, k** Time-lapse images (**j**) and kymograph analysis (**k**) showing that the GFP-STL2 and tdt-CesA6-labeled SmaCC/MASC were directly derived from Golgi through a membrane tail-stretching process. Arrows indicated the Golgi, while the arrowheads indicated the end of the Golgi tail in (**j**). The progress of the Golgi membrane tail-stretching events at the indicated time points was schematically presented above the images of (**j**). Bars = 2 μm.

basal cells of etiolated hypocotyls, GFP-STL2 colocalized with tdTomato-CesA6 in Golgi-derived SmaCCs/MASCs (Supplementary Fig. 4a–d and Supplementary Movie 5). CesA abundance at the PM fluctuates during the diurnal cycle in seedlings grown in short photoperiods (4 h day/20 h night) on sucrose-free media[35]. Here, the CesA density at the PM is high during the end of the day and this is reversed at the end of the night with high levels of CesAs accumulating in

SmaCCs/MASCs. Time-lapse imaging revealed that these SmaCCs/MASCs were also derived from Golgi (Supplementary Fig. 4e–h and Supplementary Movie 6). Because the SmaCCs/MASCs were formed in the same manner in all the three instances outlined above, if not specified, we use epidermal cells in the basal region of etiolated hypocotyls as a model to study how CSCs were relayed from Golgi to SmaCCs/MASCs.

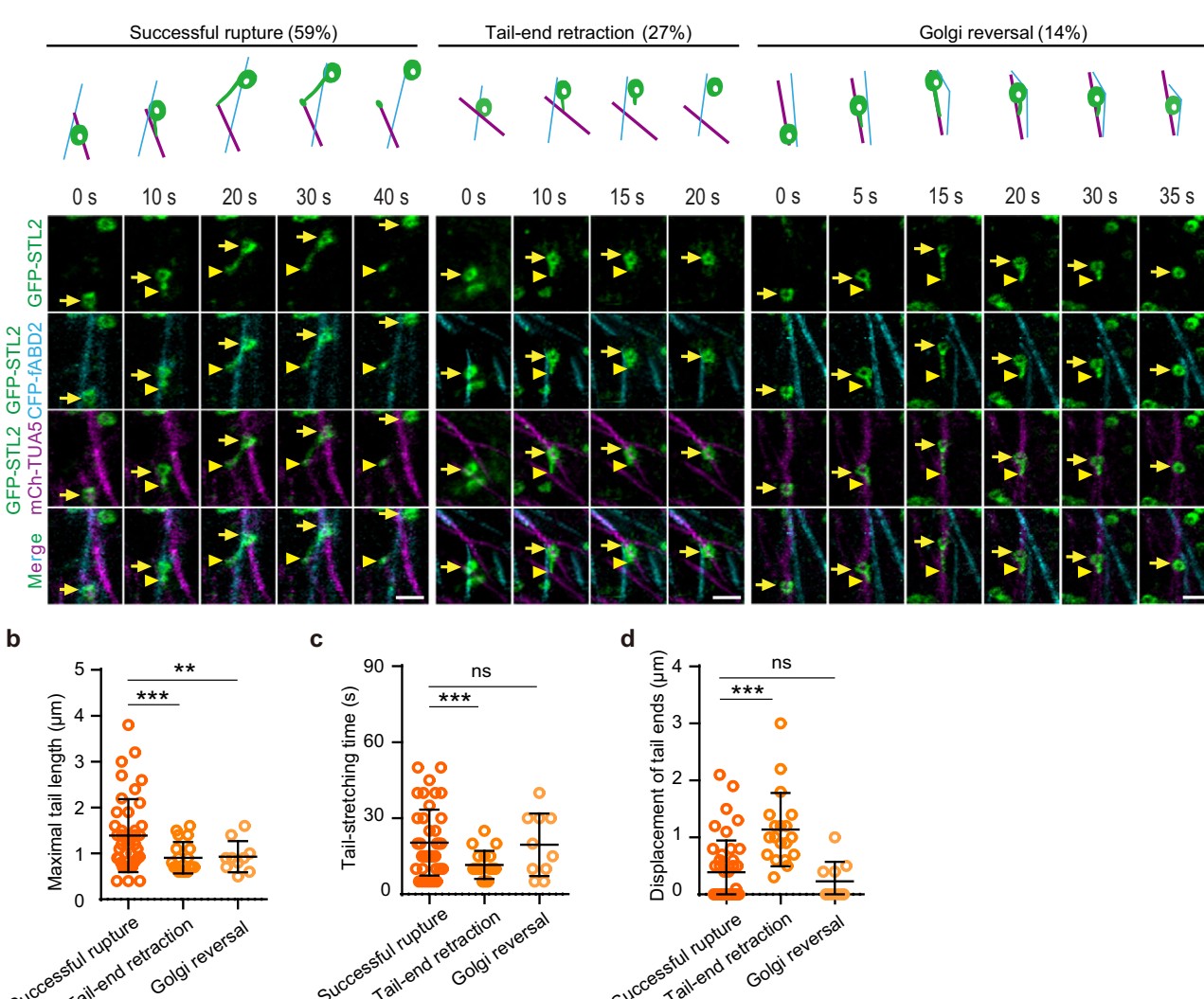

**Fig. 2 | Actin filaments and microtubules were involved in the formation of Golgi-derived SmaCCs/MASCs. a** Time-lapse images showing the spatiotemporal interaction among GFP-STL2-localized Golgi, CFP-fABD2-labeled actin filaments and mCherry-TUA5-labeled cortical microtubules during the Golgi membrane tail-stretching processes. The Golgi moved along actin filaments and parts of the Golgi membrane protruded to form a long tail when it encountered and engaged with nearby microtubules. The left panels showed that in 59% of the membrane tail-stretching events, successful rupture of the tail was achieved through efficient forward movement of the Golgi and anchor of the tail end at microtubules (referred to as successful rupture). The fission of the tail generated a SmaCC/MASC at microtubules (40 s). The middle panels showed that in 27% of the membrane tail-stretching events, detachment of the tail end from microtubules resulted in retraction of the tail back to Golgi (10–20 s; referred to as tail-end retraction). The right panels showed that 14% of the membrane tail-stretching events were followed by the reversed movement of the Golgi, which led to back-pedaling of the tail to Golgi (15–35 s; referred to as Golgi reversal). Arrows indicated the Golgi, while the arrowheads indicated the end of the Golgi tail. The progress of the Golgi membrane tail-stretching events at the indicated time points was schematically presented above the images. Bars = 2 µm. **b–d** Quantification of the maximal tail length (**b**), the tail stretching time (**c**) and the displacement of the tail ends (**d**) during the Golgi membrane tail-stretching processes as exemplified in (**a**). Values are mean ± SD. In total, 71 membrane tail-stretching events in 8 cells were quantified. **P value < 0.01; ***P value < 0.001; ns, not significant; two-sided Student's t test.

## Actin filaments and microtubules contribute to the formation of Golgi-derived SmaCCs/MASCs

Because Golgi moves along actin filaments and SmaCCs/MASCs localize at microtubules[2,3,32,33,36,37], we next examined the spatiotemporal interaction among GFP-STL2-labeled Golgi and SmaCCs/MASCs with microtubules and actin filaments. We generated triple fluorescent plants expressing mCherry-TUA5, CFP-fABD2 (for actin filament visualization) and GFP-STL2. Time-lapse imaging revealed that GFP-STL2 fluorescent Golgi moved along actin filaments throughout the cytosol (Fig. 2a). During the "membrane tail-stretching" process, the Golgi encountered and engaged with nearby microtubules (Fig. 2a). This engagement appeared to anchor the membrane to the microtubule and further movement of the

Golgi along the actin filament led to the formation of a membrane tail attached to the microtubule (Fig. 2a and Supplementary Movie 7). Interestingly, the end of the membrane tail was not always stationary at one position on microtubules for the whole "membrane tail-stretching" process. It sometimes moved along microtubules for some distance before becoming stationary (left panel in Fig. 2a) or detached microtubules after the membrane tail was stretched (middle panel in Fig. 2a). Only around 59% (n = 42/71 events in 8 cells) of the Golgi tail-anchoring/membrane-stretching events were followed by membrane fission, thus generating a microtubule-associated SmaCC/MASC (left panel in Fig. 2a and Supplementary Movie 7; referred to as successful rupture). In the remaining instances, the tail-anchoring/membrane-stretching was

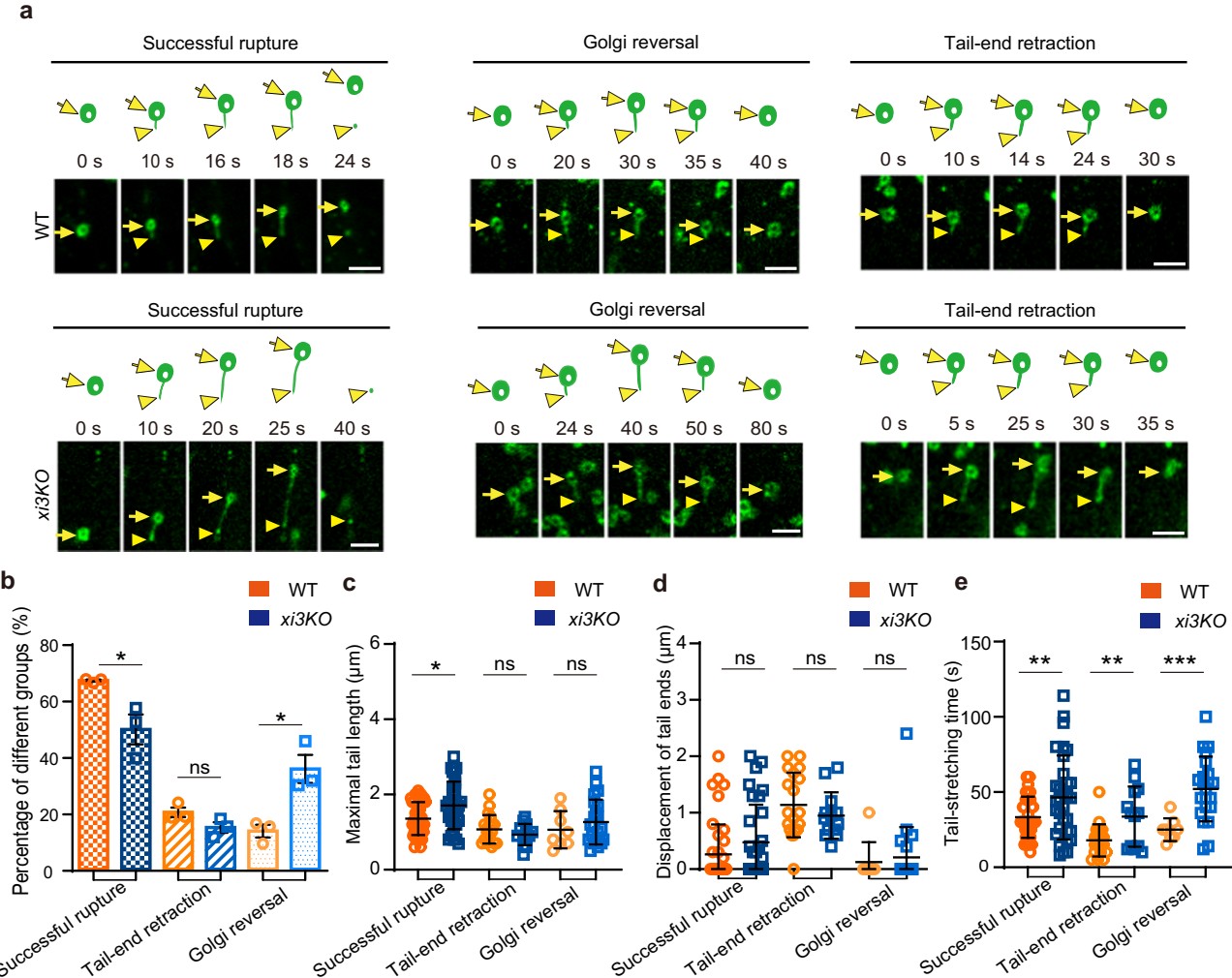

**Fig. 3 | The Golgi membrane tail-stretching processes were impaired in myosin *xik xi1 xi2* triple mutants (*xi3KO*). a** Time-lapse images showing the Golgi membrane tail-stretching processes labeled by GFP-STL2 in the control and *xi3KO* cells. Representative events of successful rupture, Golgi reversal and tail-end retraction were shown in the left, middle and right panels, respectively. Arrows indicated the Golgi, while the arrowheads indicated the end of the Golgi tail. The progress of the membrane tail-stretching events at the indicated time points was schematically presented above the images. Bars = 2 μm. **b** Percentage of different groups of Golgi membrane tail-stretching events in the control and *xi3KO* cells.

Values are mean ± SE. *n* = 24 cells from 12 WT seedlings examined over 3 independent experiments; *n* = 27 cells from 12 *xi3KO* seedlings examined over 3 independent experiments. *$P$ value < 0.05; ns, not significant; two-sided Student's *t* test. **c**–**e** Quantification of the maximal tail length (**c**), the displacement of the tail ends (**d**) and the tail stretching time (**e**) during the Golgi membrane tail-stretching processes as exemplified in (**a**). Values are mean ± SD. In total, 70 membrane tail-stretching events in 11 control cells and 67 membrane tail-stretching events in 10 *xi3KO* cells were quantified. *$P$ value < 0.05; **$P$ value < 0.01; ***$P$ value < 0.001; ns, not significant; two-sided Student's *t* test.

followed either by the detachment of the membrane from the microtubule (27%; *n* = 19/71 events; middle panel in Fig. 2a; Supplementary Movie 8; referred to as tail-end retraction) or reversed Golgi movement (14%; *n* = 10/71 events; right panel in Fig. 2a and Supplementary Movie 9; referred to as Golgi reversal), both failing to yield SmaCCs/MASCs (Fig. 2a). Insufficient stretching of the membrane tails led to a significantly reduced maximal tail length in both the tail-end retraction group and the Golgi reversal group compared to the successful rupture group (Fig. 2b). Quantification analysis further revealed that unsuccessful anchoring of the membrane to the microtubule led to shortened tail-stretching time, but significantly increased displacement of the tail end in the tail-end retraction group compared to the successful rupture group (Fig. 2c, d). Taken together, these data suggest that both efficient forward movement of Golgi along actin filaments and tight anchoring of Golgi membranes at microtubules are important for the formation of Golgi-derived SmaCCs/MASCs.

## Myosin-dependent Golgi movement facilitates the formation of Golgi-derived SmaCCs/MASCs

The trafficking of Golgi stacks is largely driven by three highly expressed myosins (XI-K, XI-1 and XI-2) in Arabidopsis[38–40]. We asked if the formation of Golgi-derived SmaCCs/MASCs was affected in the previously characterized *myosin xik xi1 xi2* triple-knockout mutant, referred to as *xi3KO*[40]. Consistent with previous reports, the overall velocity of Golgi movement was significantly reduced in *xi3KO* cells compared to that of control cells (Supplementary Fig. 5a). The reduced Golgi movement in the *xi3KO* did not significantly alter the rate of tail end retraction behavior of the Golgi, but markedly increased the frequency of Golgi reversal compared to the control cells (Fig. 3a, b). These observations indicate a role of myosin proteins in providing sufficient motive force for Golgi to continue movement despite membrane-anchoring to the microtubules. Interestingly, we observed significantly increased maximal membrane-tail length in the successful rupture groups in *xi3KO*

cells compared to the control (Fig. 3c and Supplementary Movie 10). This phenotype mainly resulted from the increased displacement of Golgi, as mutation of the myosins had minor impact on the displacement of the tail ends (Fig. 3d and Supplementary Movie 10). Although the Golgi membrane extension rate was reduced in *xi3KO* cells compared to the control (Supplementary Fig. 5b), the membrane-tail stretching time was significantly extended in the mutants (Fig. 3e). These data indicated that the speed of the Golgi movement was reduced, and therefore increased membrane-tail stretching time, in the mutants, which might be beneficial for generating longer Golgi membrane tails. Consistent with our observations in the *xi3KO* mutants, application of Pentabromopseudilin (PBP), a potent chemical that inhibits the activity of myosin proteins[27], also led to increased frequency of Golgi reversal, elongated maximal membrane tail length in the successful rupture group and elevated membrane tail-stretching time in all three groups (Supplementary Fig. 5c–g). Thus, myosin-dependent Golgi movement was required for the stretching and rupture of Golgi membrane tails, which underpin the formation of Golgi-derived SmaCCs/MASCs.

## CSI1/POM2 is essential to anchor Golgi membrane to cortical microtubules

CSI1/POM2 is necessary for the formation of isoxaben-induced SmaCCs/MASCs[34]. We therefore asked if CSI1/POM2 also contributes to the formation of Golgi-derived SmaCCs/MASCs. For this purpose, we introgressed GFP-STL2 into the previously described mCherry-CSI1/POM2 line and *csi1-3* mutant, respectively[41,42]. Colocalization analysis revealed that mCherry-CSI1/POM2 did not localize at the Golgi apparatus, but colocalized well with GFP-STL2 at SmaCCs/MASCs (Fig. 4a, b). Time-lapse imaging further showed co-localization of mCherry-CSI1/POM2 and GFP-STL2 at the membrane tail ends in 78% of the tail-stretching events (*n* = 47/60 events; Fig. 4c). The existence of CSI1/POM2 at the tail ends sometimes occurred already at the initial phase of the membrane tail-stretching processes (Fig. 4c; 0–25 s in the successful rupture group labeled by yellow arrows and arrowheads; Supplementary Movie 11; 0–24 s in the Golgi reversal group). Alternatively, the membrane tails could be stretched without apparent fluorescent CSI1/POM2 at their ends (Fig. 4c; 20–25 s in the successful rupture group labeled by white arrows and arrowheads; Supplementary Movie 11). In this latter case, the membrane tail end slid for some distance and became stationary when it encountered CSI1/POM2 (Fig. 4c; 45–50 s in the successful rupture group labeled by white arrows and arrowheads; Supplementary Movie 11). The sliding behavior of the membrane tail end was most likely along microtubules as observed in Fig. 2a (left panel, 10–20 s). In the remaining 22% of membrane tail-stretching events (*n* = 13/60 events), we did not detect mCherry-CSI1/POM2 signals at the membrane tail ends, perhaps indicating that CSI1/POM2 may not be essential for initiating the membrane tail-stretching events (Fig. 4c; tail-end retraction group; Supplementary Movie 12).

To investigate the role of CSI1/POM2 in the membrane tail-stretching process, we compared such events with mCherry-CSI1/POM2 at the membrane tail ends with those without mCherry-CSI1/POM2, and finally those in *csi1-3* mutants. The presence of CSI1/POM2 at the tail ends fully abolished the retraction of membrane tail ends, and either led to the generation of SmaCCs/MASCs (77%, *n* = 36/47 events; Fig. 4c, d), or was followed by the reversed Golgi movement to retract the tails (23%, *n* = 11/47 events; Fig. 4c, d). By contrast, when CSI1/POM2 was not present at the membrane tail ends, all the tail-stretching events ended up with membrane tail end retraction (*n* = 13/13 events; Fig. 4c, d). Similarly, while the Golgi membrane tail-stretching events were observed in *csi1-3* mutants, all of the events were followed by the membrane tail end retraction (*n* = 35/35 events; Fig. 4d, e and Supplementary Movie 13). The maximal membrane tail

length, the displacement of the tail ends and the membrane tail-stretching time were all similarly altered when CSI1/POM2 was absent from the membrane tail ends and those in *csi1-3* mutants, compared to the tails with CSI1/POM2 (Fig. 4f–h). In line with these data, STL2-labeled SmaCCs/MASCs were rarely observed in *csi1-3* mutants (Fig. 4i, j). Because CSI1/POM2 can interact with both microtubules and CesAs[16], these observations together suggested that CSI1/POM2 may enhance the anchoring of Golgi membrane tail ends to microtubules via the CesAs, which was crucial for the formation of Golgi-derived SmaCCs/MASCs.

We next determined the spatiotemporal relationship between CSI1/POM2 and cortical microtubules, using plants expressing functional 3×YFP-CSI1/POM2 and mCherry-TUA5[42]. We first applied the microtubule inhibitor oryzalin to substantially depolymerize microtubules, and the inhibitor was subsequently washed out to allow microtubule recovery. Interestingly, we observed that cortical microtubules were re-assembled at positions devoid of 3×YFP-CSI1/POM2 signals (Supplementary Fig. 6a; 300 min indicated by yellow arrow; 355 min indicated by white arrow; Supplementary Movie 14). The 3×YFP-CSI1/POM2 was subsequently recruited to newly assembled microtubules (Supplementary Fig. 6a, b and Supplementary Movie 14). These data indicated that CSI1/POM2 localizes to cortical microtubules, but that the newly assembled cortical microtubules are assembled in positions where CSI1/POM2 is absent.

## SmaCCs/MASCs may act as a transfer station on the way of CSCs from Golgi to the PM

SmaCCs/MASCs can deliver CSCs to the PM in cells grown under non-stressed conditions[33]. To determine if this holds true for the Golgi-derived SmaCCs/MASCs, time-lapse imaging was applied to analyze single CSC insertion events for the SmaCCs/MASCs labeled with both GFP-STL2 and tdTomato-CesA6. We observed successive split of one SmaCC/MASC into two components and the break-up product without GFP-STL2 fluorescence showed the slow and steady trajectories of active CSCs (Fig. 5a, b and Supplementary Movie 15). These observations support that the Golgi-derived SmaCCs/MASCs could deliver CSCs to the PM.

To further assess the influence of Golgi-derived SmaCCs/MASCs on CSC delivery, we quantified the density of CSCs at PM in *stl1 stl2* and *csi1-3* epidermal cells in the basal region of etiolated hypocotyls, respectively. The density of SmaCCs/MASCs labeled with GFP-CesA3 was markedly reduced in *stl1 stl2* mutants compared to the control (Fig. 5c, d). The density of PM-localized CSCs was also significantly decreased in the *stl1 stl2* mutant cells as compared to the control (Fig. 5e, f). Similarly, less PM-localized CSCs were observed in the *csi1-3* cells than the control (Fig. 5g, h). These data supported that the Golgi-derived SmaCCs/MASCs contributed to transfer CSCs from the Golgi to the PM, at least in cells with slow growth or when growth had ceased.

Exocytosis is responsible for the final step of CSC secretion to the PM[25,28]. We next asked if the Golgi-derived SmaCCs/MASCs acted ahead of exocytosis, in term of their roles in CSC secretion. If this is true, one would expect to observe accumulation of Golgi-derived SmaCCs/MASCs in mutants with reduced exocytosis. We therefore analyzed the density of GFP-STL2-labeled SmaCCs/MASCs in *xi3KO* epidermal cells in the apical region of etiolated hypocotyls, where the tethering/fusion of the CSC-containing vesicles to the PM is impaired[28]. Indeed, while STL2-labeled SmaCCs/MASCs were rarely detected in the actively elongating WT cells, the amount of these SmaCCs/MASCs was markedly increased in the *xi3KO* cells (Fig. 5i, j). Similarly, PBP treatment resulted in significantly increased density of STL2-labeled SmaCCs/MASCs in the epidermal cells in the apical region of etiolated hypocotyls compared to the control (Fig. 5i, j). These data suggested that the Golgi-derived SmaCCs/MASCs likely act as an intermediate station ahead of delivery of the CSCs from the Golgi to the PM.

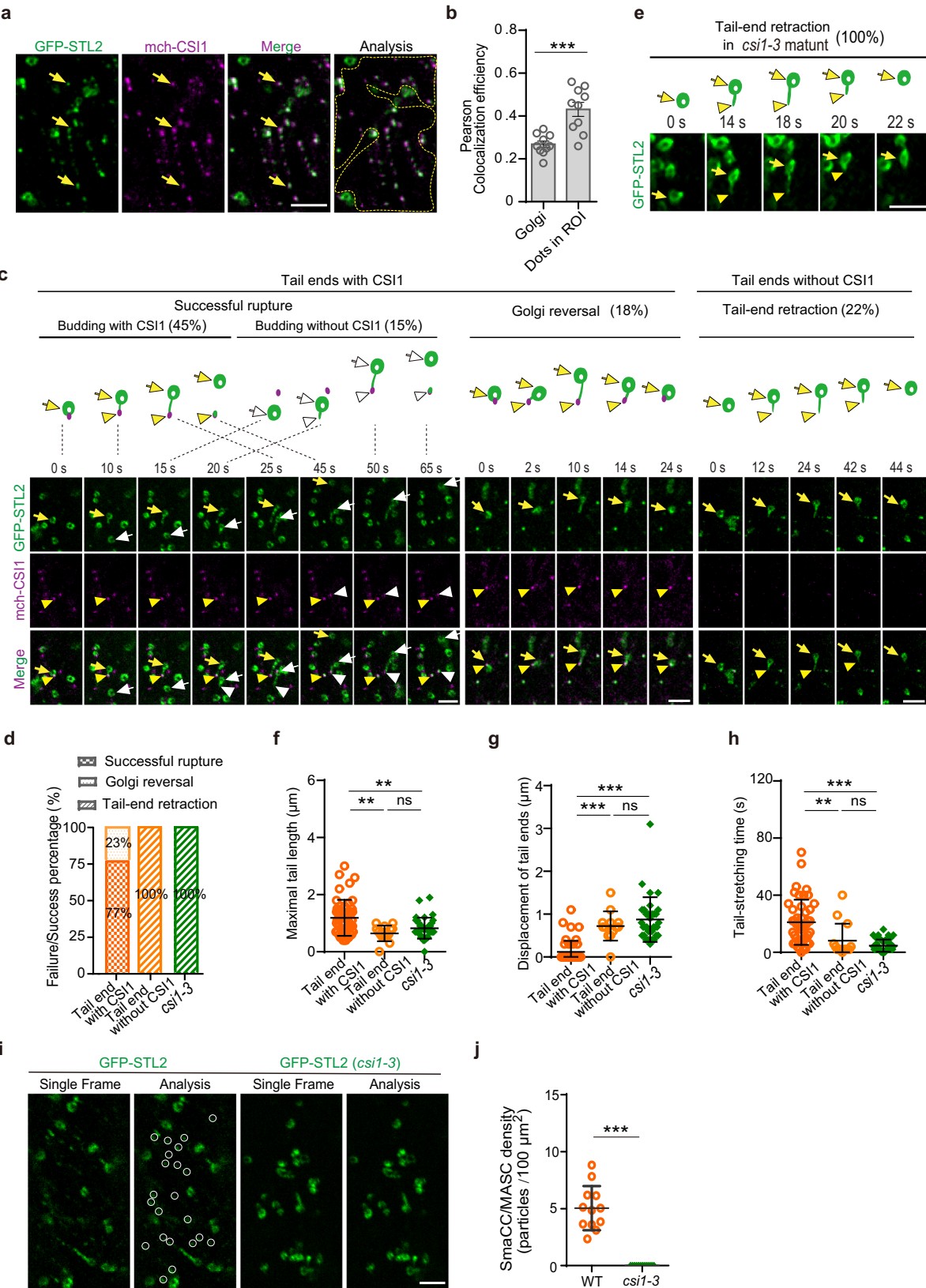

## Discussion

As one of the key structural components of the plant cell wall, cellulose plays crucial roles in plant growth and development and is also widely used as raw material for a range of industries. Because CSCs are only active at the PM, efficient secretion of them to the PM target sites is critical for cellulose synthesis in plant cells[13]. The CSCs are assembled in the Golgi, from where they are trafficked and delivered to the PM[13]. However, how the CSCs are released from the Golgi has remained unclear. SmaCCs/MASCs are CSC-containing compartments that coincide with cortical microtubules and some Golgi bodies were

**Fig. 4 | CSI1/POM2 was required for the formation of Golgi-derived SmaCCs/MASCs. a** Representative images of an epidermal cell expressing both GFP-STL2 and mCherry-CSI1/POM2 in the basal region of 4-day-old etiolated hypocotyls. Arrows indicated colocalization of the two proteins at the SmaCCs/MASCs. For analysis, region of interest (ROI) that excluded the Golgi apparatuses was marked by yellow dotted outlines. Bar = 3 μm. **b** Colocalization analysis of GFP-STL2 and mCherry-CSI1/POM2 in the Golgi and ROI as indicated in (**a**) using the Pearson correlation coefficient. Values are mean ± SD. $n = 10$ cells from 4 seedlings; ***$P$ value < 0.001; two-sided Student's $t$ test. **c** Time-lapse images showing the Golgi membrane tail-stretching processes labeled by GFP-STL2 and mCherry-CSI1/POM2. Representative events of successful rupture, Golgi reversal and tail-end retraction were showed in the left, middle and right panels, respectively. The successful rupture events included two subgroups: a group with mCherry-CSI1/POM2 signal existing already at the initial phase of the membrane tail-stretching processes (labeled by yellow arrow and arrowheads), and the other group without CSI1/POM2 at the tail end at the beginning of membrane tail-stretching processes (labeled by white arrow and arrowheads). Arrows indicated the Golgi, while the arrowheads indicated the ends of the Golgi tails. The progress of the membrane tail-stretching events at the indicated time points was schematically presented above the images.

The dashed lines in the left panel indicated the time point for which the progress was schematically shown. Bars = 3 μm. **d** Frequency of different groups of Golgi membrane tail-stretching processes in the control cells with CSI1/POM2 at the tail ends ($n = 47$ events), without CSI1/POM2 at the tail ends ($n = 13$ events), and in $csi1-3$ mutants ($n = 35$ events). **e** Time-lapse images showing the Golgi membrane tail-stretching processes labeled by GFP-STL2 in $csi1-3$ mutants. Note that all the Golgi membrane tail-stretching processes ended up as tail-end retraction in $csi1-3$ mutants. n = 35 events in 9 cells. Bar = 3 μm. **f–h** Quantification of the maximal tail length (**f**), the displacement of the tail ends (**g**) and the tail stretching time (**h**) in the control cells with CSI1/POM2 at the tail ends, without CSI1/POM2 at the tail ends, and in $csi1-3$ mutants. Values are mean ± SD. In total, 47 membrane tail-stretching events with CSI1/POM2 at the tail ends, 13 membrane tail-stretching events without CSI1/POM2 at the tail ends, and 35 membrane tail-stretching events in $csi1-3$ mutants were quantified. **$P$ value < 0.01; ***$P$ value < 0.001; ns, not significant; One-way ANOVA. **i** Representative images showing the density of GFP-STL2-localized SmaCCs/MASCs in the control and $csi1-3$ cells. Bar = 3 μm. **j** Quantification of the density of GFP-STL2-localized SmaCCs/MASCs in the control and $csi1-3$ cells. Values are mean ± SD. $n = 12$ cells from 4 WT seedlings; $n = 13$ cells from 4 $csi1-3$ seedlings. ***$P$ value < 0.001; two-sided Student's $t$ test.

observed to associate with SmaCCs/MASCs[32,33]. SmaCCs/MASCs consist of a heterogenous compartment population, which has obscured research progress on them[13,32,33]. Here, we report that STLs can specifically label a sub-population of SmaCCs/MASCs that are derived from Golgi apparatuses. We took advantage of this specificity to investigate how CSCs are relayed from Golgi to the SmaCCs/MASCs. Based on data from live-cell imaging, genetics and pharmacological experiments, we propose an anchor-pulling model of Golgi membrane tail-stretching and breaking process (Fig. 6). Firstly, microtubules can be assembled at cell cortex, where they recruit CSI1/POM2. Golgi bodies move along actin filaments and are distributed throughout the cell in a myosin-dependent manner. Secondly, the Golgi encounters cortical microtubules, with which they engage, and part of the Golgi membrane then stretches out in a membrane tail with the tail end tethered to the microtubule. Thirdly, the Golgi bodies keep moving forward along actin filaments: a process that elongate the membrane tails. The membrane tails are supported by CSI1/POM2, likely through the anchoring of CSI1/POM2 to CesAs and the microtubules. In the last steps, the tails are severed, leaving SmaCCs/MASCs associated with cortical microtubules. These SmaCCs/MASCs can further split and contribute to CSC delivery to the PM.

Actin cytoskeleton plays central roles in organelle and vesicle trafficking. In plant cells, Golgi, mitochondria and peroxisome move along actin filaments and their motions are driven by myosin activity[43,44]. Disruption of actin cytoskeleton, either by applying actin depolymerizing drugs, or mutation of *ACTIN2* and *ACTIN7* genes, leads to aggregation of Golgi and uneven distribution of CSCs at the PM. This uneven distribution coincides with underlying Golgi bodies[26,32,33]. These reports imply that CesA distribution at the PM relies on actin cytoskeleton-based distribution of Golgi in the cytoplasm. Our data provide at least two layers of evidence supporting this notion. We observe that Golgi needs to be in close contact with cortical microtubules to adhere to them to drive the formation of Golgi-derived SmaCCs/MASCs. A functional actin cytoskeleton would thus drive cell-wide distribution of Golgi to increase Golgi encountering cortical microtubules. Given that at least a subset of SmaCCs/MASCs can deliver CSCs to the PM[33], the direct transfer of CSCs between Golgi and SmaCCs/MASCs would facilitate the spatiotemporal correlation between Golgi and CSCs in PM. Recent studies have revealed that Myosin XIK, most likely via its globular tail domain (GTD), participates in the exocytosis of CSCs in plant cells[27,28]. Myosin XIK GTD interacts directly with exocyst subunits and fluorescently tagged Myosin XIK transiently colocalized with stationary foci of exocyst subunits at the PM. Pharmacological and genetic inhibition of Myosin XI activity resulted in reduced rate of appearance and lifetime of stationary

exocyst complexes at the PM, which led to impaired tethering of CSC-containing vesicles and reduced CSC secretion[28]. These studies and our work suggest that myosin XIs are involved in CSC trafficking and delivery at multiple levels, including spreading Golgi to facilitate contact with cortical microtubules, propelling Golgi movement to facilitate Golgi tail stretching for the formation of Golgi-derived SmaCCs/MASCs, and direct interaction with the exocyst complex to facilitate the tethering of CSC-containing vesicles.

The direction of cellulose microfibrils frequently coincides with the direction of the cortical microtubule array, which gave rise to the microtubule-microfibril alignment hypothesis[45–47]. The hypothesis is strongly supported by the observations that fluorescently tagged CesAs in the PM track along cortical microtubules in plant cells and the identification of the major linker protein CSI1/POM2[15,16]. CSI1/POM2 is a multiple-domain protein that can bind to both microtubules and CesAs. CSI1/POM2 tracks with CSCs at the PM and is essential for the association of CSCs with microtubules[15,16]. Apart from the function in connecting CesAs and cortical microtubules, CSI1/POM2 plays a role in PATROL1 and exocyst complex-mediated CSC secretion[25]. CSI1/POM2 marks the docking sites along cortical microtubules, which allows access of CSC-containing vesicles to cortical microtubules and ensures the association of CSCs with microtubules[25]. Moreover, CSI1/POM2 localizes at SmaCCs/MASCs, most likely through its C2 domain, and is important for the formation of stress-induced SmaCCs/MASCs[34]. Our work provides evidences that CSI1/POM2 is also required for the formation of Golgi-derived SmaCCs/MASCs under non-stressed conditions. The observation that Golgi membrane tails could be stretched out without CSI1/POM2 suggest that CSI1/POM2 may not be necessary for the initiation of Golgi deformation. It is plausible that an uncharacterized protein may locate at microtubules and is responsible for the initiation of Golgi tail stretching. Nevertheless, CSI1/POM2 is essential for anchoring the tail ends at microtubules and thus ensures the localization of newly formed SmaCCs/MASCs on microtubules. Our work, together with previous reports, support that the connection between CSCs and cortical microtubules is established via CSI1/POM2 at cell cortex when CSC-containing compartments approach the PM. The homolog of CSI1/POM2, CSI3, also binds with CesAs and displays colocalization with CesAs in the PM and SmaCCs/MASCs[48]. However, CSI3 has only minor impact on the co-alignment of CSCs and microtubules at the PM and does not impair the formation of isoxaben-induced SmaCCs/MASCs, although it is required for normal velocity of CSCs in the PM[34,48]. It therefore appears unlikely that CSI3 contributes substantially to the formation of Golgi-derived SmaCCs/MASCs.

Stress-induced SmaCCs/MASCs play a recycling role in CSC trafficking[34]. When cells are challenged with osmotic stress, salt stress

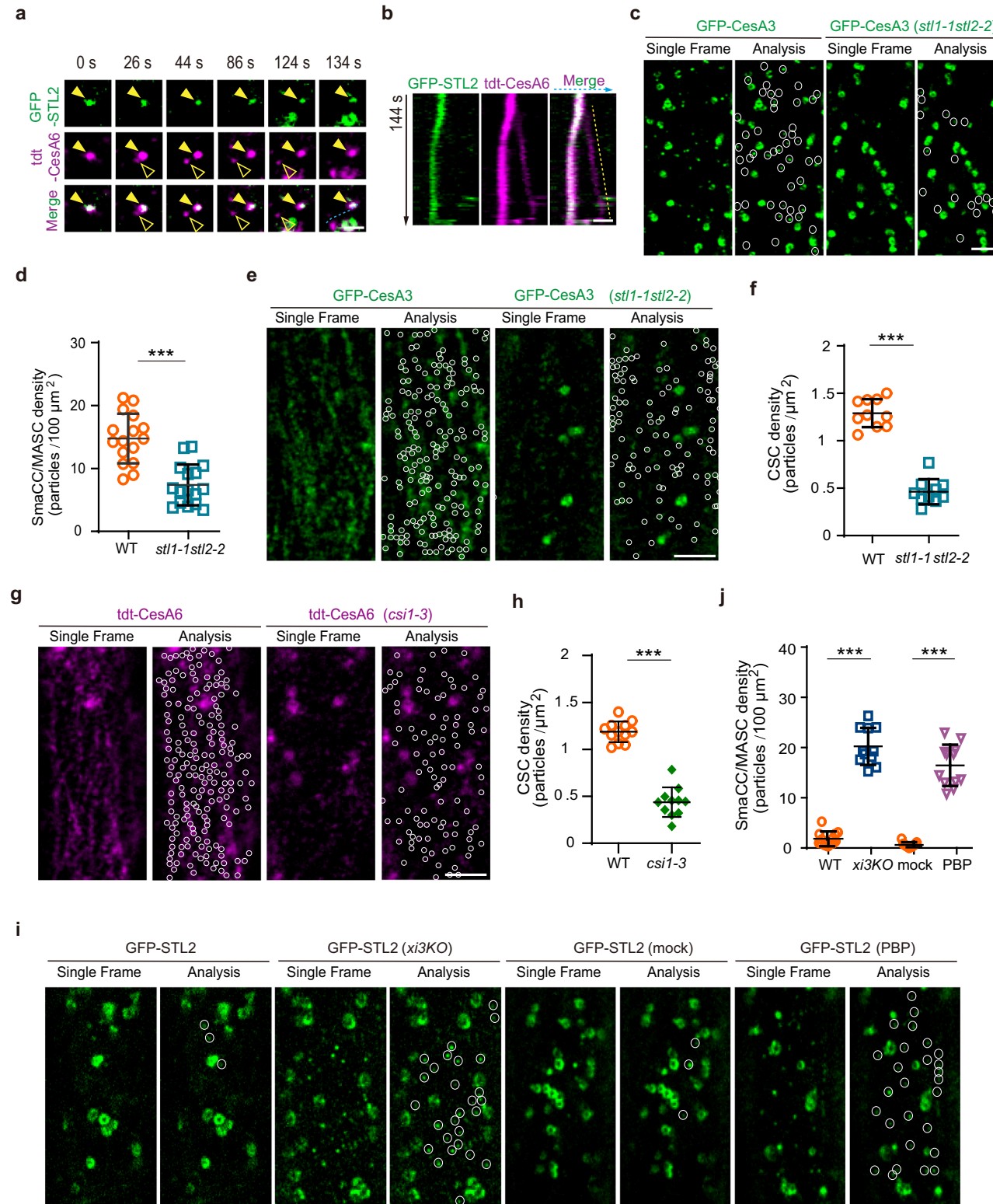

or cellulose synthase inhibitors, CSCs are depleted from the PM and are temporally stored in the SmaCCs/MASCs[32–34,49]. The CSCs are recycled back to the PM after stress exposure. In *csi1/pom2* mutants, accumulation of SmaCCs/MASCs is reduced after isoxaben treatment, which leads to significantly delayed recovery of CSCs at the PM after inhibitor washout[34]. SmaCCs/MASCs are proposed to consist of a heterogenous population of compartments[32,33]. Consistently, our data support that the Golgi-derived SmaCCs/MASCs are different than the stress-induced SmaCCs/MASCs. Our data indicate that the Golgi-

derived SmaCCs/MASCs may serve as a transfer station of CSCs on their way to the PM. Indeed, previous reports and our data observe that SmaCCs/MASCs can deliver CSCs to the PM in cells grown under non-stressed conditions[33]. In accordance with these observations, the CSC density at PM is reduced in *stl1 stl2* and *csi1-3* epidermal cells that have ceased to grow, and which lack the Golgi-derived SmaCCs/MASCs. If SmaCCs/MASCs act as an important intermediate station for CSC delivery to the PM, why do we only rarely observe such SmaCCs/MASCs in elongating cells? One possible explanation would be that

**Fig. 5 | Analysis of the dynamics of Golgi-derived SmaCCs/MASCs, the density of SmaCCs/MASCs and PM-localized CSCs in *stl1 stl2*, *csi1* and *xi3KO* mutants.**
**a**, **b** Time-lapse images (**a**) and kymograph analysis (**b**) showing single CSC insertion events for the SmaCCs/MASCs labeled with GFP-STL2 and tdTomato-CesA6. Bar = 1 µm. **c** Representative images displaying the density of GFP-CesA3-labeled SmaCCs/MASCs in control and *stl1-1 stl2-2* cells in the basal region of 4-day-old etiolated hypocotyls. Bar = 3 µm. **d** Quantification of the density of GFP-CesA3-labeled SmaCCs/MASCs as shown in (**c**). Values are mean ± SD. *n* = 16 cells from 7 seedlings for each genotype. ***P value < 0.001; two-sided Student's *t* test. **e** Representative images showing the density of GFP-CesA3 at the PM in the control and *stl1-1stl2-2* epidermal cells in the basal region of 4-day-old etiolated hypocotyls. Bar = 3 µm. **f** Quantification of the density of GFP-CesA3 at the PM as shown in (**e**). Values are mean ± SD. *n* = 10 cells from 4 seedlings for each genotype. ***P value < 0.001; two-

sided Student's *t* test. **g** Representative images showing the density of tdTomato-CesA6 at the PM in the control, and *csi1-3* epidermal cells in the basal region of 4-day-old etiolated hypocotyls. Bar = 3 µm. **h** Quantification of the density of tdTomato-CesA6 at the PM as shown in (**g**). Values are mean ± SD. *n* = 12 cells from 4 WT seedlings; *n* = 11 cells from 5 *csi1-3* seedlings. ***P value < 0.001; two-sided Student's *t* test. **i** Representative images showing the density of GFP-STL2-localized SmaCCs/MASCs at cell cortex in the control, *xi3KO*, mock-treated and PBP-treated epidermal cells in the top region of 4-day-old etiolated hypocotyls. Bars = 3 µm. **j** Quantification of the density of GFP-STL2-localized SmaCCs/MASCs in the control and *xi3KO* cells, mock-treated and PBP-treated epidermal cells as exemplified in (**i**). Values are mean ± SD. *n* = 14 cells from 5 WT seedlings; *n* = 11 cells from 5 *xi3KO* seedlings; *n* = 11 cells from 5 mock-treated seedlings; *n* = 12 cells from 5 PBP-treated seedlings. ***P value < 0.001; two-sided Student's *t* test.

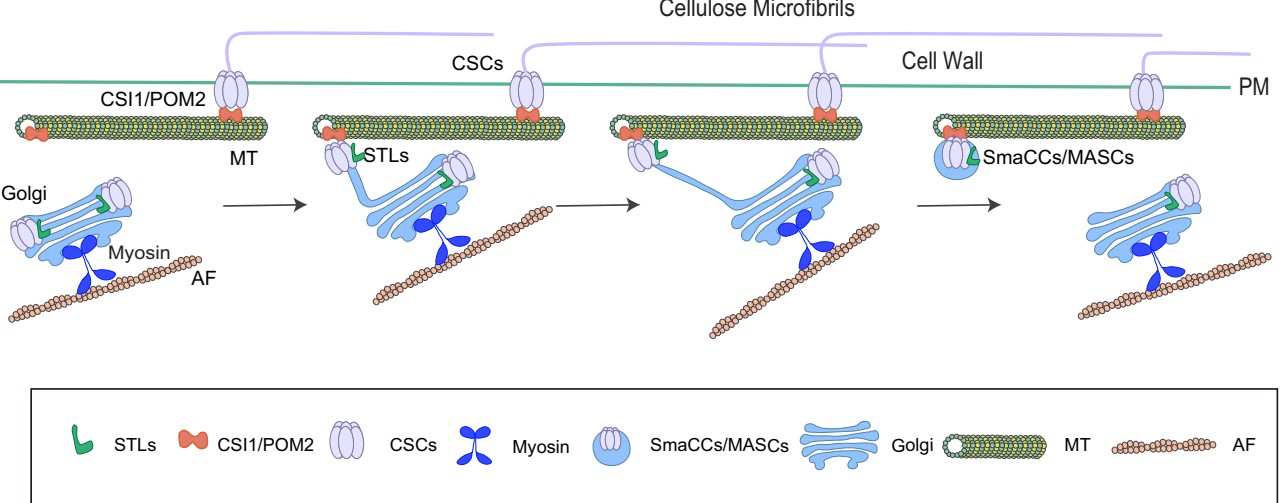

**Fig. 6 | The proposed working model of the formation of Golgi-derived SmaCCs/MASCs.** Firstly, myosin proteins propel the Golgi to move along actin filaments and microtubules are polymerized to recruit CSI1/POM2 at cell cortex. Secondly, when the Golgi encounters cortical microtubules, with which they engage, part of the Golgi membrane stretches out a membrane tail with the tail end tethered to the microtubules. CSI1/POM2 enhances the anchoring of the Golgi tail ends at microtubules. Thirdly, the Golgi continues to move along actin filaments, which elongates the Golgi tail. Finally, the Golgi tail is ruptured, generating a SmaCC/MASC at cortical microtubules.

CSCs transit such SmaCCs/MASCs quickly in the rapidly growing cells. Indeed, during the final steps of CSC delivery, exocytosis is inhibited in myosin-deficient cells. Here, SmaCCs/MASCs accumulate at the cell cortex also in cells that rapidly elongate. It is therefore likely that the exocytosis in the slow-growing cells is not as active as that in the actively growing cells, which might lead to the increased density of Golgi-derived SmaCCs/MASCs in the slow-growing cells. CesAs have been detected in proteomic and electron microscopy analysis of the TGN[32,50]. The strongly impaired CesA secretion rate in the V-ATPase mutant *det3* with abnormal pH homeostasis in TGN/EE[24], further support that CSCs may also transit TGN on their way from Golgi to the PM. How these two routes cooperate in the delivery of CSCs would be interesting questions for further investigation.

In summary, actomyosin and CSI1/POM2 cooperate in the Golgi membrane tail-stretching and breaking process for directly transferring cellulose synthase from Golgi to cortical microtubules. This non-canonical route provides insight into intracellular trafficking routes in eukaryote cells. Further study on the function of STLs and other players in this route will provide a more detailed picture of how the route is regulated.

## Methods
### Plant growth conditions and treatment
For dark growth, the surface-sterilized seeds were sowed and grown on vertical plates containing half-strength Murashige and Skoog (MS)

media supplemented with 1% sucrose and 0.5% Gelzan (Sigma-Aldrich, G1910) at pH 5.8 in the dark at 22 °C. For carbon limitation treatment, seedlings were grown on half MS media with 0.5% Gelzan (without sucrose) at pH 5.8 in short day condition (4 h day/20 h dark) for 7 days.

For inhibitor treatment, 4-day-old dark grown seedlings were treated with 100 nM isoxaben (Sigma-Aldrich, 36138) for 2 h or 0.75 µM PBP (Adipogen, BVT-0441) for 15 min, respectively[27,33]. For the microtubule recovering experiment, 4-day-old dark grown seedlings were treated with 20 µM oryzalin (Supelco, PS-410) for 8 h to fully depolymerize microtubules. The seedlings were then transferred to water to let microtubules to recover.

### Plant materials
T-DNA insertional lines for *csi1-3* (SALK_138584) were obtained from NASC (http://arabidopsis.info/) and described previously[34,41]. The mutants of *xi3KO*, *stl1-1 stl2-2* and *stl1-2 stl2-1*, the fluorescent plants of pUb10::STL1-mCherry, pCesA6::tdTomato-CesA6, pCesA3::GFP-CesA3, the native promoter-driven mCherry-CSI1/POM2 and 3×YFP-CSI1/POM2, and the marker lines of mCherry-TUA5, SYP61-CFP, mCherry-SYP32 were described previously[9,26,33,40,42,48,51–54]. The various dual and triple fluorescent lines were generated via crossing and the progeny was analyzed. The pSTL2::GFP-STL2 line in *stl1-1 stl2-2* mutant background was crossed with *csi1-3* and *xi3KO*, respectively. The line expressing pCesA6::tdTomato-CesA6 in a *prc1-1* mutant background was crossed with the *csi1-3* mutant and the line expressing

pCesA3::GFP-CesA3 in a *je5* mutant background was crossed with the *stl1-1 stl2-2* double mutant. The segregating plants were used for analyzing the SmaCCs/MASCs and CSCs in control and mutants.

## Constructs

The primers used for producing constructs are listed in Supplemental Table 1. For pSTL2::GFP-STL2 construct, the GFP6 fragment in pMDC43 was first replaced by EGFP with restriction enzymes SpeI and Asc1. The 2 × 35S promoter in pMDC43 was then replaced by the 2495 bp STL2 upstream sequence, which included the 5'-UTR and was amplified via primers pSTL2_for /pSTL2_rev, with restriction enzymes Hind3 and Kpn1. The ORF of STL2 was amplified from Col-0 cDNA via primers STL2_for/STL2_rev and cloned into pENTR using the pENTR/ D-TOPO Cloning Kit (Invitrogen, USA). The error-free construct of STL2-pENTR was then used for LR reactions of the Gateway cloning system (Invitrogen) with the modified vector of pSTL2::EGFP-pMDC43 to generate pSTL2::GFP-STL2 construct. The construct was transformed into *Agrobacterium tumefaciens* strain GV3101. The transformation of *stl1-1 stl2-2* mutants was performed by floral dip and transformants were selected by antibiotic resistance and fluorescence intensity.

For p35S::fABD2-CFP construct, the fABD2 fragment was amplified from Col-0 cDNA via primers fABD2_for/ fABD2_rev. The fragment was cloned into the pDNOR207 vector, which was further recombined into pEarleyGate102 vector by gateway methods[55]. The construct was then transformed into the Col-0 background.

## Live-cell imaging

Live-cell imaging was done essentially as described by Dai et al. 2022[56]. If not specified, live-cell imaging was performed in the epidermal cells in the bottom region (3-5 cells above the junction of hypocotyl and root) of 4-day-old etiolated hypocotyls. Time-lapse imaging of the Golgi membrane tail-stretching processes in the control, *xi3KO* and PBP-treated cells, and the interaction among GFP-STL2, tdTomato-CesA6, mCherry-TUA5 and fABD2-CFP were performed with a spinning disk confocal microscope (UltraView VoX, PerkinElmer, UK) equipped with a Nikon TiE inverted microscope, a Yokogawa Nipkow CSU-X1 spinning disk scanner, ×100 1.49 NA oil immersion objective and Hamamatsu EMCCD 9100-13. For observation of the Golgi membrane tail-stretching processes in petiole epidermal cells, seedlings were grown in half-strength Murashige and Skoog (MS) media supplemented with 1% sucrose and 0.5% Gelzan (Sigma-Aldrich, G1910) at pH 5.8 under a 16-h day/8-h night regime (22 °C) for 10 days. The first pair of rosette leaves were cut and cells from the basal region of the petioles were imaged. The time intervals were 2 s and the total duration were between 2 and 5 min.

The colocalization of GFP-STL2 and mCherry-CSI1/POM2 and the Golgi membrane tail-stretching processes in *csi1-3* mutants were visualized with a high-speed laser confocal living cell workstation (Dragonfly, Andor, Oxford Instruments, UK) equipped with a ×60 1.4 NA oil immersion objective and a Zyla 4.2 Megapixel sCMOS camera (Andor Technology). The 440-, 488- and 561-nm lasers were used for exciting the fluorescence of CFP, GFP and mCherry/tdTomato, respectively. The time intervals were 2 or 5 s and the total durations were between 3 and 10 min.

To examine if the Golgi-derived SmaCCs/MASCs acted ahead of exocytosis in term of their roles in CSC secretion, the 3-5 epidermal cells under the hook were imaged for the density of GFP-STL2-labeled SmaCCs/MASCs in the top region of *xi3KO* etiolated hypocotyls, where the tethering/fusion of the CSC-containing vesicles to the PM is impaired[28]. For the microtubule recovering experiment, time-lapse imaging was performed with intervals of 1 min for 8 h after transferring the seedlings from oryzalin solution into water. All the microscopy experiments were repeated for at least three times with multiple cells and seedlings.

## Image analysis

The acquired images were processed by Volocity (Perkin Elmer) and ImageJ (https://imagej.nih.gov/ij/). Co-localization of STL2 with CesA6, SYP61 and CSI1/POM2 was measured using the Coloc 2 plugin of ImageJ. Pearson's and Mander's coefficients were measured using the Colocalization (Coloc2) plugins in ImageJ. The kymograph analysis was performed with the Multiple Kymograph plugin in ImageJ. The density of SmaCCs/MASCs and PM-localized CSC particles were manually determined using the analyze particles tool in ImageJ. The Golgi velocity was determined using the particle tracker plugins in ImageJ. The bar graph and box diagram were made using Prism 5 (GraphPad software).

## Statistical analysis

All the representative images were obtained from three repeated experiments with similar results. The two-sided Student's *t* test and one-way ANOVA were used to analyze significant differences between groups. $*P < 0.05$, $**P < 0.01$, and $***P < 0.001$.

## Reporting summary

Further information on research design is available in the Nature Portfolio Reporting Summary linked to this article.

## Data availability

Biological materials can be obtained upon request. Source data are provided with this paper.

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

## Acknowledgements

We acknowledge the helpful discussion with Professor Paul Dupree (Cambridge University) at the initial stages of the project. We acknowledge the experimental technology center for life sciences, Beijing Normal University, and thank Dr. Xiaoyan Zhang for technical support. We acknowledge the assistance of Imaging Core Facility of Protein Research Center for Technology Development for assistance of using the high-speed laser confocal living cell workstation. This work was supported by grants from the National Natural Science Foundation of China (32070194, 32270350 and 31870174 to Y.Z., and 32100279 to T.W.). S.P. acknowledges the financial aid of Villum Investigator (Project ID: 25915), DNRF Chair (DNRF155) and Novo Nordisk Laureate (NNF19OC0056076), Novo Nordisk Emerging Investigator (NNF20OC0060564); Novo Nordisk Data Science (NNF0068884) and Lundbeck foundation (Experiment grant, R346-2020-1546) grants.

## Author contributions

Y.Z. and S.P. initiated the project. Y.Z., L.L., and T.W. designed the experiments; L.L., T.W., Y.B., and P.Y. performed the experiments; P.D. generated the CFP-fABD2-expressing plants; L.L., T.W., and L.D. analyzed the data; Y.Z., L.L., T.W., and S.P. wrote, reviewed, and edited the manuscript with input from all of the authors. Y.Z. supervised the project.

## Competing interests

The authors declare no competing interests.
