## [Peer Review File · Nature Communications]

Actomyosin and CSI1/POM2 cooperate to deliver cellulose synthase from Golgi to cortical microtubules in ArabidopsisReviewer #1 (Remarks to the Author):

This is a really impressive piece of work. How the cellulose synthase complex is trafficked to the plasma membrane is an important question. The authors have provided a very detailed framework for at least one of the pathways required for transport of intact cellulose synthase complexes to the plasma membrane. They are able to show that Golgi-derived vesicles are dependent upon the formation of a Golgi tubule and independent of the trans-Golgi network. The formation of the tubule is dependent upon both movement of the Golgi via myosin motors on actin filaments and microtubules that appear to mark the sites of vesicle attachment that is required to successfully separate the vesicle from the Golgi tubule. The work focuses on delivery of complexes under normal, unstressed growth as opposed to stress-based recycling of the complex, but nevertheless this is an important pathway that has been clearly elucidated by the authors.

In general I find the study to be thoroughly carried out with proper quantitation of the important events. It also utilises several different mutant backgrounds meaning this represents a substantial body of work.

There is only one part of the interpretation that I am unclear on. Why are Golgi tubules longer in when myosin-based Golgi movement is prevented? The authors suggest that maybe because they are moving more slowly. This may be so, but in this case the rate of tubule formation should be much slower. I am not sure if the authors have measured rate directly. From the images it is hard to say. From the available data, it should be possible for the authors to measure the rate of tubule extension. If it is slower, then the interpretation stands up, if it is not slower then they need a rethink!

Other minor points.

Why has no one reported the tubules before? Obviously STL has not been particularly extensively studied, but CESA proteins have been extensively studied for live cell imaging. Have they been observed with no one recognising the significance of them? I note that the endogenous promoter was used for STL, was this also true for CESA6?

Could the author clarify the level of colocalisation between STL and CESA6. While they seem to label the same Golgi and/or vesicle at high resolution they seem almost mutually exclusive. This is particularly clear in the movie file.

Line 269, 78% how does that relate to figure 4.

I am slightly unsure about how convincing the additional Golgi marker data is. The SYP32 labelling appears clear, but the sialyltransferase is less convincing. There is only a single still of rather clustered Golgi and the movie is similar. How does this compare to a randomly selected still with STL or CESA? There must be many images with no Golgi tails apparent. I would suggest that this either needs better quantitation or be left out, I do not consider it necessary.

Reviewer #2 (Remarks to the Author):

The noteworthy results in this study by Liu et al. is the discovery of a novel exit mechanism for cellulose synthase (CESA) enzymes from the Golgi during plant cell wall synthesis. With high-quality live-cell imaging, they demonstrate that the Golgi-localized STELLO proteins co-localize with a subset of small cellulose synthase (CESA) compartments (SmaCCs/MASCs), and they describe the cytoskeletal mechanisms required to generate these Golgi-derived SmaCCs. The evidence for the 'membrane tail-stretching' events as a non-canonical exit route from Golgi is strong: it is specific to STL and CESA, and is not observed in other Golgi/TGN markers such as ST, SYP32, and SYP61. They show the tail formation is altered in myosin mutants whose Golgi are slower, and that the microtubule-binding and cellulose-interacting protein CSI1 is required for the process, anchoring the membrane tail to microtubules. The quality of the images, the image analysis, and data interpretation are very high quality. The claim presented in the Discussion are all supported by the data and illustrated in Figure 6: microtubule assembly and CSI1 recruitment, Golgi encountering cortical microtubules and stretching the membrane in a tail tethered by CSI1, this process requiring the actin-myosin cytoskeleton, finally membrane fission and MASCs on

cortical microtubules.

One concern regarding significance to the field is that the phenomenon of membrane tail stretching and the generation of STL-labelled SmaCCs is found only under specific circumstances. It is observed in low carbon conditions and in the base of the Arabidopsis dark-grown hypocotyl (three-five cells above the junction of the hypocotyl and root). It is not a common feature of elongating cells, where cellulose production is most active. The Discussion in lines 348-349 implies that this work addresses this gap in our understanding: "how the CSC are released from the Golgi remains unclear". How CSC are released from Golgi in elongating cells where cellulose synthesis is critical is still unclear. This begs the question: why are the STL-CESA SmaCCs abundant in these conditions? The authors address this in the final paragraph, acknowledging that they can't reconcile a TGN-exit from the STL-SmaCC routes, and leave the matter for further investigation.

On this note, in the methods, it says "The 3-5 epidermal cells under the hook were imaged for the density of GFP-STL2-labelled SmaCCs/MASCs in the top region of xi3KO etiolated hypocotyls." Presumably this refers to the data in Figure 3. Why could STL2-labeled Golgi membrane tail-stretching processes be observed in the top of the hypocotyl in this experiment but not the others?

The one claim in this work that is not well-supported is the question of whether the STL-tagged SmaCCs generated by Golgi membrane tails fuse with the cell membrane to deliver CSC to the cell surface. The evidence for this is circumstantial, as *csi1* mutants have fewer SmaCCs and lower PM signal of CESA6, while (relying heavily on the Zhang et al., 2021 paper) SmaCC/MASCs accumulate in the cortex of myosin mutants. If this data supports an "important role of the Golgi-derived SmaCCs/MASCs in transferring CSCs from the Golgi to the PM", it would be good to analyze single CSC insertion events in the *stello* double mutants. The clear analysis done in the Zhang et al., 2021 paper, where they follow the erratic movement, tethering/docking, and then steady movement phases of CSC insertion provides a guide. A key unanswered question is what happens to the STL during CSC exocytosis? This is also left ambiguously in the model in Figure 6. If the final destination of the SmaCCs/MASCs is not the cell membrane, then are these structures part of a downregulation of CSC at the cessation of growth?

Minor points

1. Supplemental Figure 2 shows that STL-SmaCCs do not overlap with stress-induced SmaCCs, but this is done in top of hypocotyl. In Supp2c, there is no control without isoxaben, so we don't know what the level of co-localization between STL and CESA in SmaCCs in the top of the hypocotyl (in Supp2a, it is only shown as 'occasionally observed at small intracellular compartments'.)
2. Line 247 states "we observed significantly increased maximal membrane-tail length in the successful rupture groups in xi3KO cells compared to the control (Fig. 3c; Supplemental Video 9). This phenotype mainly resulted from the increased displacement of Golgi,..." Since Supplemental Figure 5a shows decreased Golgi velocity in the mutants, why would there be increased displacement? I think that the conclusion on line 258 should be that myosin-dependent Golgi movement is required for the rupture but not stretching of the tails.
3. Figure 4d has typo in legend 'matunt'
4. In Figures 2b-d and Figures 4f-h, Student t-tests are done for multiple pair-wise comparisons. This should be corrected or a different test applied.
5. The methods don't specify if the CESA6 lines used to create the double tdTomato
 1. (tdt)-CesaA6 and GFP-STL2 were in a mutant (*prc1-1*) or wild-type background. If the wild-type copy of CESA6 is present in these lines, could it influence the results?
 2. Line 387- typo 'exocystst'
 3. I note that the authors use the work 'important' ten times in the manuscript. It becomes somewhat meaningless and overblown, especially when the claim is not well supported, e.g. "These data supported an important role of the Golgi-derived SmaCCs/MASCs in transferring CSCs from the Golgi to the PM"

Point-by-point responses to Reviewers' comments

We appreciate the insightful comments and suggestions from the Reviewers. We have carefully performed the suggested experiments and revised the manuscript. All changes have been highlighted in RED font color in the revised manuscript and explained below.

REVIEWER COMMENTS

Reviewer #1 (Remarks to the Author):

This is a really impressive piece of work. How the cellulose synthase complex is trafficked to the plasma membrane is an important question. The authors have provided a very detailed framework for at least one of the pathways required for transport of intact cellulose synthase complexes to the plasma membrane. They are able to show that Golgi-derived vesicles are dependent upon the formation of a Golgi tubule and independent of the trans-Golgi network. The formation of the tubule is dependent upon both movement of the Golgi via myosin motors on actin filaments and microtubules that appear to mark the sites of vesicle attachment that is required to successfully separate the vesicle from the Golgi tubule. The work focuses on delivery of complexes under normal, unstressed growth as opposed to stress-based recycling of the complex, but nevertheless this is an important pathway that has been clearly elucidated by the authors.

In general I find the study to be thoroughly carried out with proper quantitation of the important events. It also utilises several different mutant backgrounds meaning this represents a substantial body of work.

R: We appreciate the Reviewer for the encouraging comments.

There is only one part of the interpretation that I am unclear on. Why are Golgi tubules longer in when myosin-based Golgi movement is prevented? The authors suggest that maybe because they are moving more slowly. This may be so, but in this case the rate of tubule formation should be much slower. I am not sure if the authors have measured rate directly. From the images it is hard to say. From the available data, it should be possible for the authors to measure the rate of tubule extension. If it is slower, then the interpretation stands up, if it is not slower then they need a rethink!

R: The rate of Golgi tubule extension has been directly measured as suggested. As shown in Supplemental Fig. 5b, this rate is significantly reduced in both myosin triple mutants (*xi3KO*) and myosin inhibitor (PBP)-treated cells compared to the control.

Other minor points.

Why has no one reported the tubules before? Obviously STL has not been particularly extensively studied, but CESA proteins have been extensively studied for live cell imaging. Have they been observed with no one recognising the significance of them?

R: We thank the Reviewer for comments and have carefully checked the literature. In the Nature Cell Biology paper from David W. Ehrhardt's lab (Gutierrez et al, 2009), the connection of Golgi

and SmaCCs/MASCs (labeled by YFP-CESA6) is described as “Some Golgi bodies were observed to associate with SmaCCs under normal conditions and to show physical tethering to microtubule-associated SmaCCs during osmotic stress or isoxaben treatment”. Similarly, in the Plant Cell paper from Samantha Vernhettes’ lab (Crowell et al, 2009), the association of Golgi with SmaCCs/MASCs (labeled by GFP-CESA3) in untreated hypocotyl cells is described as “Detailed analysis in untreated cells in the hypocotyl revealed that Golgi pauses occurred in proximity to MASCs and could last as long as 73 s (n = 34, mean = 22 s)”. Thus, the Golgi tubules have been observed with FP-CESAs, but the phenomenon was interpreted as connection between Golgi and SmaCCs/MASCs. An introduction on the previous observations has been added in the revised manuscript (Lines 121-122).

I note that the endogenous promoter was used for STL, was this also true for CESA6?

R: Yes, the endogenous promoter was used for tdTomato-CESA6 as previously described (Sánchez-Rodríguez et al., 2012; Sampathkumar et al., 2013). This point has been clarified in the revised manuscript (Lines 505-506) and the references have been added in the revised manuscript (Lines 730-736).

Could the author clarify the level of colocalisation between STL and CESA6. While they seem to label the same Golgi and/or vesicle at high resolution they seem almost mutually exclusive. This is particularly clear in the movie file.

R: This is a technical issue for imaging two fluorophores associated with the highly motile and/or constant wiggling Golgi apparatus and/or vesicles with the spinning disc microscope equipped with one EMCCD. The fluorescence of GFP and tdTomato was sequentially collected with a lag of about 500 ms in our settings, which led to the movies that STL and CESA6 seem mutually exclusive in Golgi at high resolution. A spinning disc microscope with dual EMCCD may be helpful to resolve this issue, but unfortunately we do not have this kind of equipment. Nevertheless, in our previous paper (Zhang et al., 2016), we detailed analyzed the localization of STLs in Golgi. As shown in the images below (Supplementary Fig. 4c in Zhang et al., 2016), quantification of

colocalization among STLs and various marker proteins support that STL proteins colocalize well with tdt-CesA6 and GFP-CesA3 in the Golgi apparatus (Zhang et al., 2016).

To further clarify the colocalization of between STLs and CesAs, we applied Myosin inhibitor Pentabromopseudilin (PBP) to inhibit the movement of Golgi. As shown in the images below,

GFP-STL2 and tdTomato (tdt)-CesA6 colocalize in Golgi and the SmaCCs/MASCs after the treatment. These observations, together with previous reports, support that STLs and CesAs are highly co-localized in the Golgi and SmaCCs/MASCs. We did not include the colocalization data after PBP treatment into the revised manuscript. However, if the Editor and Reviewer insist that we include these results, we are of course open for such suggestions.

Line 269, 78% how does that relate to figure 4.

R: 78% is the percentage of Golgi tail ends with mCherry-CSI1/POM2, including the three groups of budding with CSI1 (45%), budding without CSI1 (15%; the colocalization of mCherry-CSI1/POM2 and GFP-STL2 happens after the budding event in this group) and Golgi reversal (18%) in Figure 4c.

I am slightly unsure about how convincing the additional Golgi marker data is. The SYP32 labelling appears clear, but the sialyltransferase is less convincing. There is only a single still of rather clustered Golgi and the movie is similar. How does this compare to a randomly selected still with STL or CESA? There must be many images with no Golgi tails apparent. I would suggest that this either needs better quantitation or be left out, I do not consider it necessary.

R: The images of sialyltransferase (ST)-RFP has been removed as suggested. We also strengthen the data using SYP32-mCherry that did not label the Golgi membrane tail-stretching events by generating double fluorescent lines expressing both SYP32-mCherry and GFP-STL2. As shown in Supplemental Fig. 3a and Supplemental Video 3, while GFP-STL2 clearly labeled the Golgi membrane tails, SYP32-mCherry did not.

Reviewer #2 (Remarks to the Author):

The noteworthy results in this study by Liu et al. is the discovery of a novel exit mechanism for cellulose synthase (CESA) enzymes from the Golgi during plant cell wall synthesis. With high-

quality live-cell imaging, they demonstrate that the Golgi-localized STELLO proteins co-localize with a subset of small cellulose synthase (CESA) compartments (SmaCCs/MASCs), and they describe the cytoskeletal mechanisms required to generate these Golgi-derived SmaCCs. The evidence for the 'membrane tail-stretching' events as a non-canonical exit route from Golgi is strong: it is specific to STL and CESA, and is not observed in other Golgi/TGN markers such as ST, SYP32, and SYP61. They show the tail formation is altered in myosin mutants whose Golgi are slower, and that the microtubule-binding and cellulose-interacting protein CSI1 is required for the process, anchoring the membrane tail to microtubules. The quality of the images, the image analysis, and data interpretation are very high quality. The claim presented in the Discussion are all supported by the data and illustrated in Figure 6: microtubule assembly and CSI1 recruitment, Golgi encountering cortical microtubules and stretching the membrane in a tail tethered by CSI1, this process requiring the actin-myosin cytoskeleton, finally membrane fission and MASCs on cortical microtubules.

R: We appreciate the Reviewer for the encouraging comments.

One concern regarding significance to the field is that the phenomenon of membrane tail stretching and the generation of STL-labelled SmaCCs is found only under specific circumstances. It is observed in low carbon conditions and in the base of the Arabidopsis dark-grown hypocotyl (three-five cells above the junction of the hypocotyl and root). It is not a common feature of elongating cells, where cellulose production is most active. The Discussion in lines 348-349 implies that this work addresses this gap in our understanding: "how the CSC are released from the Golgi remains unclear". How CSC are released from Golgi in elongating cells where cellulose synthesis is critical is still unclear. This begs the question: why are the STL-CESA SmaCCs abundant in these conditions? The authors address this in the final paragraph, acknowledging that they can't reconcile a TGN-exit from the STL-SmaCC routes, and leave the matter for further investigation.

R: We thank the Reviewer for the comments. We have observed similar phenomenon of Golgi membrane tail stretching and the generation of Golgi-derived SmaCCs/MASCs in the epidermal cells of light-grown leaf petioles (Supplementary Fig. 4a-d), which have been used as another cell model to observe CSC dynamics (Duncombe et al., 2022). Thus, while this phenomenon is less commonly seen in the elongating hypocotyl cells, it is certainly biologically relevant in cells that have stopped growing or that grow slowly. We have also performed additional experiments showing that the Golgi-derived SmaCCs/MASCs can split and deliver CSCs to the plasma membrane (Figure 5a,b). Thus, the STL-labeled SmaCCs contribute to the delivery of CSC from Golgi to the plasma membrane at least in the slow growing or fully elongated cells. As discussed in the manuscript, interesting questions certainly remain to be addressed, e.g., how the TGN-exit route and the Golgi-SmaCC routes cooperate in the delivery of CSCs in different cells (Lines 470-471).

On this note, in the methods, it says "The 3-5 epidermal cells under the hook were imaged for the density of GFP-STL2-labelled SmaCCs/MASCs in the top region of *xi3KO* etiolated hypocotyls." Presumably this refers to the data in Figure 3. Why could STL2-labeled Golgi membrane tail-stretching processes be observed in the top of the hypocotyl in this experiment but not the others?

R: This description of methods refers to the data in Figure 5i,j and has been clarified in the revised manuscript (Lines 553-557). The data in Figure 3 were obtained from the epidermal cells in the basal region of etiolated hypocotyls.

In Figure 5i-j, we attempt to examine if the Golgi-derived SmaCCs/MASCs acted ahead of exocytosis, in terms of their roles in CSC secretion. If this is true, one would expect to observe accumulation of Golgi-derived SmaCCs/MASCs in mutants with reduced exocytosis. We therefore analyzed the density of GFP-STL2-labelled SmaCCs/MASCs in *xi3KO* epidermal cells in the apical region of etiolated hypocotyls, where the tethering/fusion of the CSC-containing vesicles to the PM is impaired (Zhang et al., 2021).

The one claim in this work that is not well-supported is the question of whether the STL-tagged SmaCCs generated by Golgi membrane tails fuse with the cell membrane to deliver CSC to the cell surface. The evidence for this is circumstantial, as *csi1* mutants have fewer SmaCCs and lower PM signal of CESA6, while (relying heavily on the Zhang et al., 2021 paper) SmaCC/MASCs accumulate in the cortex of myosin mutants. If this data supports an “important role of the Golgi-derived SmaCCs/MASCs in transferring CSCs from the Golgi to the PM”, it would be good to analyze single CSC insertion events in the *stello* double mutants. The clear analysis done in the Zhang et al., 2021 paper, where they follow the erratic movement, tethering/docking, and then steady movement phases of CSC insertion provides a guide. A key unanswered question is what happens to the STL during CSC exocytosis? This is also left ambiguously in the model in Figure 6. If the final destination of the SmaCCs/MASCs is not the cell membrane, then are these structures part of a downregulation of CSC at the cessation of growth?

R: We thank the Reviewer for the comments and suggestions. We have analyzed the formation process of Golgi-derived SmaCCs/MASCs in *st1 st2* double mutants. As shown in the images below, the Golgi membrane-tail deformation or extension is markedly inhibited in *st1 st2* mutants, which leads to failure of generating Golgi-derived SmaCCs/MASCs. We did not include these data into the revised manuscript, as we prefer to use STLs as specific markers to study the roles of cytoskeleton and related proteins in generating Golgi-derived SmaCCs/MASCs in this manuscript. The mechanism underlying the roles of STLs in Golgi membrane deformation and/or extension would be a follow-up story. However, again, if the Editor and Reviewer insist that we include these results, we are of course open for such suggestions.

Based on these observations, we think the SmaCCs/MASCs observed in the *stl1 stl2* mutants were likely derived from the plasma membrane or other source, but not from Golgi. We thus did not track the destination of these SmaCCs/MASCs in the *stl1 stl2* double mutants. Instead, we tracked the behavior of SmaCCs/MASCs labeled by both STL2-GFP and tdT-CESA6, which were derived from Golgi. As shown in the Figure 5a,b, the Golgi-derived SmaCC/MASC can split into two components and the break-up product without GFP-STL2 fluorescence showed the slow and steady trajectories of active CSCs (Fig. 5a,b). These observations, together with the data that the density of PM-localized CSCs was reduced in *csi1-3* and *stl1 stl2* mutants, support that the Golgi-derived SmaCCs/MASCs contribute to CSC delivery.

Taken together, these data support that the plasma membrane is at least one of the destinations of Golgi-derived SmaCCs/MASCs.

Minor points

1. Supplemental Figure 2 shows that STL-SmaCCs do not overlap with stress-induced SmaCCs, but this is done in top of hypocotyl. In Supp2c, there is no control without isoxaben, so we don't know what the level of co-localization between STL and CESA in SmaCCs in the top of the hypocotyl (in Supp2a, it is only shown as 'occasionally observed at small intracellular compartments'.)

R: The control has been added in Supplemental Fig. 2b-c. In the control, there are abundant CSCs at the PM, which make it challenging to distinguish between the SmaCCs/MASCs and CSCs with the fluorescence of tdTomato-CesA6. Nevertheless, we observe that the GFP-STL2-labeled SmaCCs/MASCs display the fluorescence of tdTomato-CesA6, indicating a colocalization.

2. Line 247 states "we observed significantly increased maximal membrane-tail length in the successful rupture groups in *xi3KO* cells compared to the control (Fig. 3c; Supplemental Video 9). This phenotype mainly resulted from the increased displacement of Golgi,..." Since Supplemental Figure 5a shows decreased Golgi velocity in the mutants, why would there be increased displacement? I think that the conclusion on line 258 should be that myosin-dependent Golgi movement is required for the rupture but not stretching of the tails.

R: As suggested by Reviewer 1, we have quantified the Golgi membrane-tail extension velocity. This velocity is significantly reduced in *xi3KO* and PBP-treated cells. Based on these data, we think myosin-dependent Golgi movement is required for both stretching of the tails and the rupture of the tails. We thus did not revise the conclusion.

The increased displacement of Golgi in *xi3KO* cells is due to the increased time for Golgi membrane-tail extension (Supplemental Fig.5b). It is likely that the rupture of the tails is impaired in *xi3KO* cells so that the extension is elongated, which leads to increased Golgi membrane tails.

3. Figure 4d has typo in legend 'matunt'

R: This has been corrected (Line 864).

4. In Figures 2b-d and Figures 4f-h, Student t-tests are done for multiple pair-wise comparisons. This should be corrected or a different test applied.

R: The comparisons have been corrected to between the groups of tail-end retraction or Golgi reversal to the group of successful rupture in Figure 2b-d. One-way ANOVA has been applied to Figure 4f-h for multiple pair-wise comparisons.

5. The methods don't specify if the CESA6 lines used to create the double tdTomato 1. (tdt)-CesA6 and GFP-STL2 were in a mutant (*prc1-1*) or wild-type background. If the wild-type copy of CESA6 is present in these lines, could it influence the results?

R: The tdt-CESA6 lines used is in the *prc1-1* mutant background as described previously (Sánchez-Rodríguez et al., 2012; Sampathkumar et al., 2013). This information and the references have been added in the revised manuscript (Lines 505-508; Lines 730-736).

2. Line 387- typo 'exocytst'

R: This has been corrected.

3. I note that the authors use the work 'important' ten times in the manuscript. It becomes somewhat meaningless and overblown, especially when the claim is not well supported, e.g. "These data supported an important role of the Golgi-derived SmaCCs/MASCs in transferring CSCs from the Golgi to the PM"

R: We thank the Reviewer for pointing out this. We have revised the word throughout the manuscript and toned down the claim to be "These data supported that the Golgi-derived SmaCCs/MASCs contributed to transfer CSCs from the Golgi to the PM, at least in cells with slow or ceased growth" in the revised manuscript (Lines 338-340).

References

Crowell EF, Bischoff V, Desprez T, Rolland A, Stierhof YD, Schumacher K, Gonneau M, Höfte H, Vernhettes S. Pausing of Golgi bodies on microtubules regulates secretion of cellulose synthase complexes in Arabidopsis. *Plant Cell*, 2009, 21(4): 1141-54.

Duncombe SG, Chethan SG, Anderson CT. Super-resolution imaging illuminates new dynamic behaviors of cellulose synthase. *Plant Cell*. 2022, 34(1): 273-286.

Gutierrez R, Lindeboom JJ, Paredes AR, Emons AM, Ehrhardt DW. Arabidopsis cortical microtubules position cellulose synthase delivery to the plasma membrane and interact with cellulose synthase trafficking compartments. *Nat Cell Biol*, 2009, 11(7): 797-806.

Sampathkumar A, Gutierrez R, McFarlane HE, Bringmann M, Lindeboom J, Emons AM, Samuels L, Ketelaar T, Ehrhardt DW, Persson S. Patterning and lifetime of plasma membrane-localized cellulose synthase is dependent on actin organization in Arabidopsis interphase cells. *Plant Physiol*, 2013, 162(2): 675-88.

Sánchez-Rodríguez C, Bauer S, Hématy K, Saxe F, Ibáñez AB, Vodermaier V, Konlechner C, Sampathkumar A, Rüggeberg M, Aichinger E, Neumetzler L, Burgert I, Somerville C, Hauser MT, Persson S. Chitinase-like1/pom-pom1 and its homolog CTL2 are glucan-interacting proteins important for cellulose biosynthesis in Arabidopsis. *Plant Cell*, 2012, 24(2): 589-607.

Zhang Y, Nikolovski N, Sorieul M, Vellosillo T, McFarlane HE, Dupree R, Kesten C, Schneider R, Driemeier C, Lathe R, Lampugnani E, Yu X, Ivakov A, Doblin MS, Mortimer JC, Brown SP, Persson S, Dupree P. Golgi-localized STELLO proteins regulate the assembly and trafficking of cellulose synthase complexes in Arabidopsis. *Nat Commun*, 2016, 7: 11656.

Reviewer #1 (Remarks to the Author):

I think the authors have done a good job in addressing the comments raised. I am happy with their response.

Reviewer #2 (Remarks to the Author):

The authors have done a good job to address all of my concerns. I further appreciate how they toned down the excessive use of "important" while still conveying the impact of their findings. I think this is an exciting addition to our understanding of cellulose synthesis.